# Representing individual electronic states for machine learning GW band structures of 2D materials

Nikolaj Rørbæk Knøsgaard [1✉] & Kristian Sommer Thygesen [1]

Choosing optimal representation methods of atomic and electronic structures is essential when machine learning properties of materials. We address the problem of representing quantum states of electrons in a solid for the purpose of machine leaning state-specific electronic properties. Specifically, we construct a fingerprint based on energy decomposed operator matrix elements (ENDOME) and radially decomposed projected density of states (RAD-PDOS), which are both obtainable from a standard density functional theory (DFT) calculation. Using such fingerprints we train a gradient boosting model on a set of 46k $G_0W_0$ quasiparticle energies. The resulting model predicts the self-energy correction of states in materials not seen by the model with a mean absolute error of 0.14 eV. By including the material's calculated dielectric constant in the fingerprint the error can be further reduced by 30%, which we find is due to an enhanced ability to learn the correlation/screening part of the self-energy. Our work paves the way for accurate estimates of quasiparticle band structures at the cost of a standard DFT calculation.

[1] Computational Atomic-scale Materials Design (CAMD), Department of Physics, Technical University of Denmark, 2800 Kgs. Lyngby, Denmark.
✉email: nirkn@dtu.dk

The electronic band structure is one of the most fundamental and important characteristics of a crystalline solid. It relates the quantum mechanical energy levels of an electron in the solid to its (crystal) momentum and provides the basis for describing and understanding a range of materials properties. As a consequence, the accurate prediction of electronic band structures represents a cornerstone problem of computational condensed matter physics.

Density functional theory (DFT)[1] with semi-local exchange-correlation functionals[2] is the standard method for solving the electronic structure problem of materials from first principles. However, the DFT single-particle energies do not in general provide an accurate model for the electronic band structure[3]. Instead, the gold standard for band structure calculations is represented by the GW self-energy method[4], which provides the true quasiparticle (QP) band structure, i.e., it goes beyond a mean-field description by explicitly accounting for exchange and many-body screening effects[5,6]. In ref. [7] the mean absolute error on the calculated bandgap relative to experimental references for a set of ten simple semiconductors and insulators was found to be 2.05 eV for DFT-LDA and 0.31 eV for non-self-consistent $G_0W_0$@LDA. Very similar results have been found in other studies[8,9]. The improved accuracy of the GW method comes at the price of a significantly more involved methodology and a much higher computational cost. In practice, this means that GW calculations are limited to small-scale studies of relatively simple materials.

Recently, machine learning (ML) has attracted widespread interest as a means to predict materials properties without performing expensive quantum mechanical calculations[10–15]. In the context of bandgap predictions, Zhou et al. trained a support vector machine on 3896 experimental bandgaps using a representation based only on elemental properties of the constituent atoms[16]. Rajan et al. used different regressions methods to predict bandgaps of MXene crystals using a training set of 76 $G_0W_0$ bandgaps and a representation encoding atomic and structural properties[17]. Liang et al. used a representation based on atomic ionicity descriptors to predict GW bandgaps of a set of 2D semiconductors[18]. In all these previous studies, the ML model was trained to predict the size of the bandgap rather than the full $k$-resolved band structure. Thereby, important information is missed including the type of the bandgap (direct or indirect), the curvature of the valence and conduction bands at the extrema points (effective masses), and the position and dispersion of other bands away from the bandgap. Predicting the full band structure directly from the atomic structure of the material is a daunting challenge that, although possible in principle, would require highly sophisticated ML models and immense amounts of training data.

Here we take a different approach, in which the output from a DFT calculation is taken as input to an ML model to predict the full GW band structure. The philosophy behind our approach is that standard DFT calculations are computationally very cheap, in particular, compared to GW, and although they do not directly produce the desired precision, they hold the gist of the material's genome and thus should provide an excellent starting point for accurate property predictions. In our scheme, the rich, but unmanageable, information contained in the DFT wave functions is encoded into low dimensional fingerprints via energy-resolved orbital projections and operator matrix elements. These state-specific electronic fingerprints provide a description of the local environment of a given electronic eigenstate in the infinite-dimensional Hilbert space and are thus analog to the well-known fingerprints used to describe atoms in chemical environments[19].

Using a data set of 286 $G_0W_0$ band structures of non-magnetic 2D semiconductors comprising a total of 46,000 ($\varepsilon_{nk}^{QP}$, $k$) pairs, we train a gradient boosting algorithm to predict the $G_0W_0$ correction of an eigenstate from its DFT fingerprint. The method achieves a mean absolute error (MAE) of 0.14 eV for individual band energies and 0.18 eV for the bandgap. These deviations are significantly smaller than the typical size of the $G_0W_0$ corrections and also lower than the accuracy of the $G_0W_0$ method itself. The model can be further and significantly improved by adding static electronic polarisability to the fingerprint. A SHAP feature analysis reveals that the inclusion of the polarisability allows the ML model to distinguish between materials with similar PBE band structures but different dielectric screening properties, which is directly related to the size of the GW correction.

We have used the resulting ML model to obtain $G_0W_0$ band structures for ~700 2D semiconductors from the Computational 2D Materials Database (C2DB)[20,21]. These materials are additional to the data set used in this study, and the band structures will be published on the C2DB web page[22].

## Results

Figure 1a shows an example of a PBE (orange) and $G_0W_0$ (green) band structure for monolayer $MoS_2$ (note that spin–orbit interactions are not included throughout this work). It is clear that there are significant differences between the two descriptions. First of all, $G_0W_0$ yields a QP bandgap of 2.53 eV in good agreement with the experimental value of 2.5 eV[23] while PBE yields a significantly smaller bandgap of 1.58 eV. It can also be noted that unoccupied bands are shifted up in energy while occupied bands are shifted down. This is in fact a general trend across all the materials in the data set and it leads to a double peak in the histogram of $G_0W_0$ corrections with the peak of negative (positive) corrections corresponding to occupied

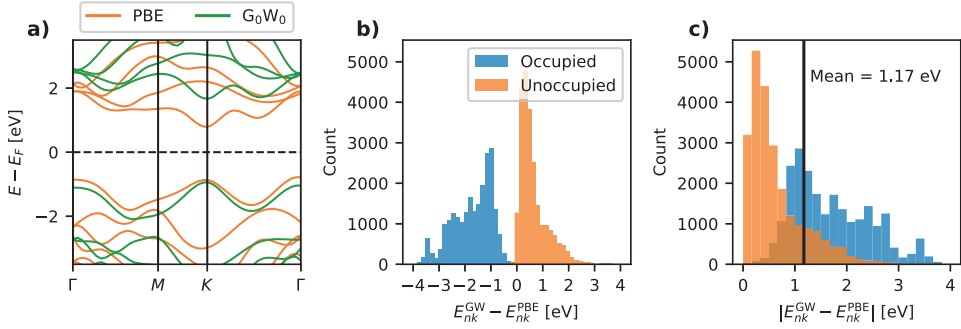

**Fig. 1 $G_0W_0$ data. a** Example of PBE and $G_0W_0$ band structures of monolayer $MoS_2$. The prediction target data is the difference in energy between the PBE and $G_0W_0$ energies. **b** Histogram of the $G_0W_0$ corrections for all states in all materials. **c** Histogram of the absolute values of the $G_0W_0$ corrections with a mean of 1.17 eV.

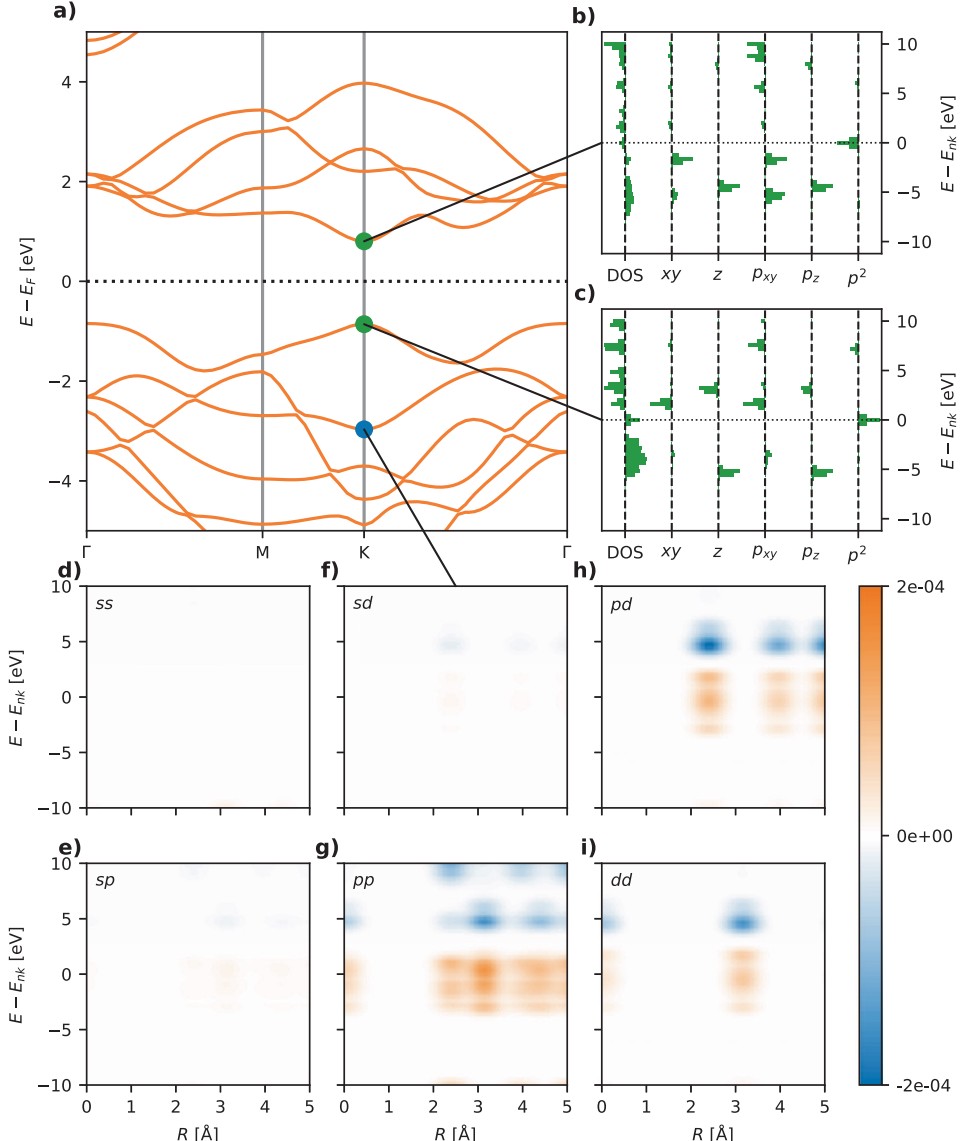

**Fig. 2 Visualization of electronic state fingerprints for MoS₂. a** Shows the PBE band structure. **b, c** Show ENDOME fingerprints of the conduction band minimum and valence band maximum states for the K-point. **d–i** Show six RAD-PDOS fingerprints for combinations of *s*, *p*, and *d* orbitals.

(empty) bands, see Fig. 1b. The absolute values of the $G_0W_0$ corrections range from 0 to 3 eV with an average value of 1.17 eV, see the histogram in Fig. 1c. Returning to the band diagram in panel (a) we further note that not all the bands are shifted by the same amount—even when disregarding the different signs for occupied/empty bands. Although for most materials, all the occupied bands experience similar, though material-specific, shifts and the same holds for the empty bands, there are several examples, like MoS₂, where this is not the case. Therefore, an accurate prediction of $G_0W_0$ corrections for general bands requires a representation that not only encodes the occupation of the state but also information about the energy and shape of the wave function and its relation to other relevant states of the crystal.

**Electronic fingerprints**. The ENDOME and RAD-PDOS representations, defined in the Methods Section, are attempts to generalize the notion of the local environment of an atom, which has been successfully employed to represent solids and molecules in machine learning studies, to the case of an electronic state. The ENDOME fingerprint represents the local environment of an energy eigenstate $|nk\rangle$ in terms of operator matrix elements between the state itself and other eigenstates of the crystal, $|\langle nk|\hat{A}|n'k'\rangle|^2$. These matrix elements are arranged on a grid as a function of the energy difference $\varepsilon_{nk} - \varepsilon_{n'k'}$, and their sign is used to encode the occupation of the final state $|n'k'\rangle$. With the ENDOME fingerprint, two states are thus considered similar if they have similar matrix elements with other states of similar relative energies. In this work, we include matrix elements for the position operator, momentum operator, and Laplacian operator. Since we exclusively consider 2D materials in the present work, the fingerprints are split into in-plane and out-of-plane components for the position operator (labeled *xy* and *z*, respectively) and the momentum operator (labeled $p_{xy}$ and $p_z$). The RAD-PDOS fingerprint is a correlation function in energy and radial distance between the atomic orbital projections (onto angular momentum channels *s*, *p*, and *d*) of the reference eigenstate and all other eigenstates of the crystal. Figure 2 visualizes the two types of fingerprints for three different electronic states of MoS₂.

Any reasonable fingerprint should comply with certain general requirements[13] of which invariance and simplicity are the most

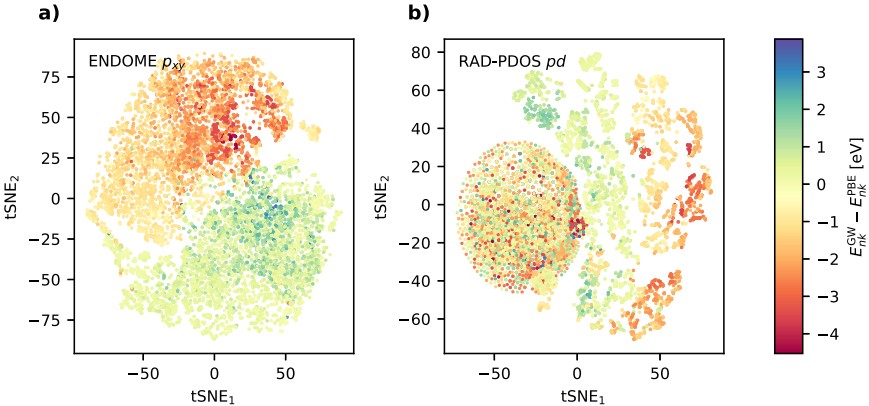

**Fig. 3 tSNE visualizations of fingerprints. a** tSNE components of ENDOME $p_{xy}$. **b** tSNE components of RAD-PDOS pd fingerprints color-coded with the GW corrections. For $p_{xy}$, states with similar GW corrections are also close in fingerprint space. In **b** a large amount of the states with both positive and negative GW corrections have similar distances in fingerprint space, corresponding to the materials without $d$-electrons where the RAD-PDOS pd fingerprint will be all zeros.

fundamental. In the present context, this means that the fingerprint should be invariant with respect to the choice of the unit cell (number of primitive cells, rotations, and translations), the gauge used for the Bloch wave functions and that it should be computationally cheap to generate compared to a full $G_0W_0$ calculation. Both the ENDOME and RAD-PDOS fingerprints clearly fulfill these requirements. Besides the invariance and simplicity conditions, the fingerprints should also be *unique* such that two different systems (here electronic states) are not mapped to the same fingerprint, and they should be *descriptive* such that systems with similar properties are close in fingerprint space. The interpretation and quantitative assessment of notions such as different systems and similar properties are obviously problem-dependent. This fact can make it difficult for problem independent fingerprints like the ENDOME and RAD-PDOS to meet these requirements in general. This is, however, not a principal problem, and can usually be solved by increasing the size of the training data set, at least as long as the fingerprints are complex and flexible enough to capture the variations in the considered systems that are relevant to the specific learning problem.

An impression of the descriptiveness of the fingerprints can be obtained from Fig. 3, which shows two-dimensional projections of the ENDOME-$p_{xy}$ and RAD-PDOS-$dd$ fingerprints using $t$-distributed stochastic neighbor embedding (tSNE) color-coded by the GW corrections. It is clear that data points, which are close in $p_{xy}$-space have similar GW corrections. The $pd$ fingerprint is also descriptive for some data points, but there is also a large blob of data points that are indistinguishable in fingerprint space but have very different GW corrections. Not unexpectedly, these points correspond to the subset of materials without valence $d$-electrons, which results in all-zero $pd$ fingerprint vectors. The tSNE plots for the other components of the ENDOME and RAD-PDOS fingerprints look similar.

**State energies**. To predict the state-specific $G_0W_0$ corrections to the PBE eigenvalues of 2D semiconductors, we use the XGBoost package[24] to build a machine learning model based on a gradient boosting algorithm for decision tree ensembles. The $G_0W_0$ data set was described and analysed in detail in ref. [25]. We split the data set into a training set of 228 randomly selected materials (37,851 electronic states) and a test set consisting of the remaining 58 materials (8766 electronic states). As an objective function, we use the mean absolute error (MAE) between the predicted and actual $G_0W_0$ corrections. The electronic states

### Table 1 Summary of results.

| Methods | Target property MAE | |
|---|---|---|
| | **Bandgap (eV)** | **State energies (eV)** |
| $G_0W_0$ vs. experiment | $\approx 0.3$ | N/A |
| PBE vs. $G_0W_0$ | 1.70 | 1.17 |
| HSE06 vs. $G_0W_0$ | 0.85 | 0.47 |
| PBE with ideal scissor-operator vs. $G_0W_0$ | 0 | 0.17 |
| ML (8VB + 4CB) vs. $G_0W_0$ | 0.23, 0.18[(*)] | 0.14, 0.11[(*)] |
| ML (VB + CB) vs. $G_0W_0$ | 0.18, 0.15[(*)] | 0.31, 0.22[(*)] |

The table shows the mean absolute error (MAE) on the bandgap and individual state energies for $G_0W_0$ versus experiments and different approximate methods versus $G_0W_0$, respectively. The MAE on state energies is always evaluated for the eight highest valence bands (VB) and four lowest conduction bands (CB). ML(X) refers to the test set MAE of the gradient boosting model after training on all bands (8VB+4CB) or only the highest valence and lowest conduction band (VB+CB), respectively. The values marked by (*) are obtained after training the model with the static polarisability of the materials included as extra features in the fingerprint.

are represented by the ENDOME and RAD-PDOS fingerprints supplemented by a set of extra features consisting of the occupation of the state ($f_{nk} = 0$, 1), its distance to the Fermi energy ($\varepsilon_{nk} - E_F$), the PBE bandgap of the material ($E_{gap}$), and the static averaged in-plane and out-of-plane polarisabilities of the material ($\frac{1}{2}(\alpha_x + \alpha_y)$ and $\alpha_z$). The averaged in-plane polarisability is used to ensure invariance of the feature with respect to rotations of the 2D material in the plane, which is important for materials with in-plane anisotropy. The effect of including the polarisabilities in the fingerprint has been analysed separately (see later discussion).

The results of the model together with relevant baselines for assessing its performance are summarized in Table 1. The first row shows the estimated accuracy of our target $G_0W_0$ data relative to experiments based on previous reports in the literature[7–9]. Experimental data for individual band/state QP energies are scarce and subject to significant uncertainties, and thus do not represent a meaningful reference. The remaining rows of the Table show the mean absolute error (MAE) on the bandgap and individual state energies for different approximate methods versus $G_0W_0$. The MAE on state energies is evaluated over all the bands for which $G_0W_0$ data is available, namely the eight highest valence bands (VB) and four lowest conduction bands (CB). The second and third rows are straightforward comparisons of band energies from PBE and HSE06 with $G_0W_0$, respectively. The fourth row shows the MAE between $G_0W_0$ and

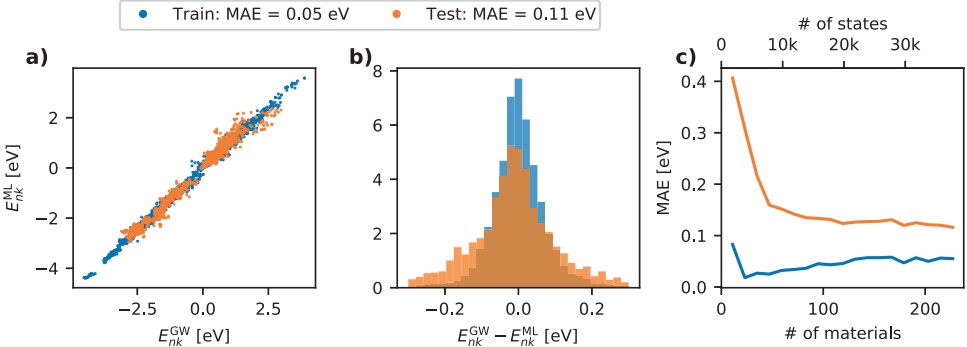

**Fig. 4 Machine learning results. a** Parity plot showing the ML predicted vs. true values of the GW correction for individual states for the train and test set. The MAEs of the train and test set are 0.05 and 0.11 eV, respectively. **b** Histograms of the prediction residuals of the train and test set. **c** Learning curve for the ML model showing validation MAE as a function of the number of materials/states in the training set.

PBE after the occupied and unoccupied PBE energies have been rigidly shifted (by applying a scissors operator) to match the valence band maximum (VBM) and conduction band minimum (CBM) of the $G_0W_0$ band structure. From this, it follows that the lowest possible MAE on individual band energies obtainable with a model trained to predict only the VBM and CBM energies is 0.17 eV. The last two rows of the table show the MAE on the test set obtained with the XGBoost model (see below for more details). Improved performance for the bandgap can be obtained by training the model only on the highest valence and lowest conduction band (last row); however, such a restriction on the training data reduces the prediction accuracy for bands further away from the bandgap. The numbers marked by (*) refer to the MAE obtained when the static polarisability of the materials is included in the fingerprint (see later discussion).

In the following, unless stated otherwise, results refer to the case where the model has been trained on all bands (8VB + 4CB) and with the static polarizabilities included in the fingerprint.

Figure 4a shows a parity plot of the predicted vs. true values for the train and test set. The evaluation yields MAEs of 0.05 and 0.11 eV for the train and test set, respectively. To test for the potential bias of the model, the residual distributions are plotted in Fig. 4b, showing that both the train and test set have residuals distributed evenly around 0 eV. To estimate the effect of adding more data to the train set, a learning curve is shown in Fig. 4c. The learning curve is calculated by continuously adding more materials to the training set while evaluating the performance on a constant test set. The test set MAE decreases significantly up to ≈50 materials after which the learning curve flattens considerably, although still presenting a slightly decreasing MAE. This suggests that a generalizable model can be trained using a rather limited number of materials, though it should be noted that overfitting issues decrease with the amount of materials in the training set. In general, it is difficult to assess whether the learning ability of the model is limited by the flexibility of the model/fingerprint or by the noise level in the data set. We do stress, however, that the numerical precision of the $G_0W_0$ corrections is not expected to be much better than 0.05 eV due to errors introduced by e.g., plane-wave extrapolation and linearisation of the self-energy, see ref. [25]. This could explain (part of) the finite prediction error of the model.

All MAEs reported in this paper were evaluated for a specific, randomly generated test set of 58 materials. We have verified that this test set is representative and fair by comparing it to MAEs obtained for 100 different random test sets, see Methods section.

The data used to train and evaluate the ML model represent states/energies evaluated at discrete uniformly distributed k-points of the Brillouin zone. However, the resulting ML model

can of course be used to predict the $G_0W_0$ energy corrections of states at arbitrary k-points and thereby generate full, densely sampled band structures. Figure 5 shows examples of ML-generated band structures for $PtO_2$, $SbClTe$, $GeS_2$, and $CaCl_2$, which are all test set materials. For comparison, the PBE and the true discrete $G_0W_0$ energies are also shown. Overall, the ML bands closely interpolate the true $G_0W_0$ energies. In cases where the ML bands deviate, e.g. the conduction bands of $CaCl_2$, they still present a better description than PBE. Interestingly, the ML model is able to deviate from a scissors operator that would ascribe the same corrections to all occupied and all unoccupied bands, respectively. This is for example clear in the $PtO_2$ band structure where the four conduction bands are shifted by different amounts. We note that the single-point regression nature of the model, i.e., the fact that the model does not explicitly couple different k-points, can sometimes lead to weak and unphysical wiggles in the machine-learned band energies. These qualitative errors may be reduced by applying a smoothing function (e.g., a Gaussian filter) as post-processing of the ML energies across bands. This has been done for the plots in Fig. 5.

**Bandgaps**. The ML state energies can be translated into ML bandgaps by simply calculating the vertical difference between conduction band minimum and valence band maximum. Figure 6 shows parity plots of the predicted bandgaps vs. $G_0W_0$ bandgaps for an ML model trained on all bands and an ML model trained only on valence and conduction bands. Due to the discreteness of the original $G_0W_0$ data, the ML bandgap has been evaluated on the same states (discrete k-points) that define the $G_0W_0$ gap. The PBE and HSE06 data are also shown as baselines. Only data from the test set has been used for the comparison. The PBE and HSE06 functionals systematically underestimate the bandgaps leading to MAEs of 1.70 and 0.85 eV, respectively. The ML model trained on all bands achieves an MAE on the bandgap of 0.18 eV, while training the ML model only on valence and conduction bands reduces the bandgap MAE to 0.15 eV, but at the cost of increasing the MAE on the individual state energies across all bands from 0.11 to 0.22 eV.

While our ML model and fingerprints allow for the prediction of state-specific properties, such as individual band energies, it is of interest to compare its accuracy on bandgap predictions to alternative schemes reported in the literature. Lee and coworkers[26] used nonlinear support vector regression with fingerprints containing the Kohn-Sham bandgap obtained with both the PBE and the mBJ xc-functionals, together with a set of features describing the constituent chemical elements, to predict $G_0W_0$ bandgaps of inorganic bulk semiconductors. Using a

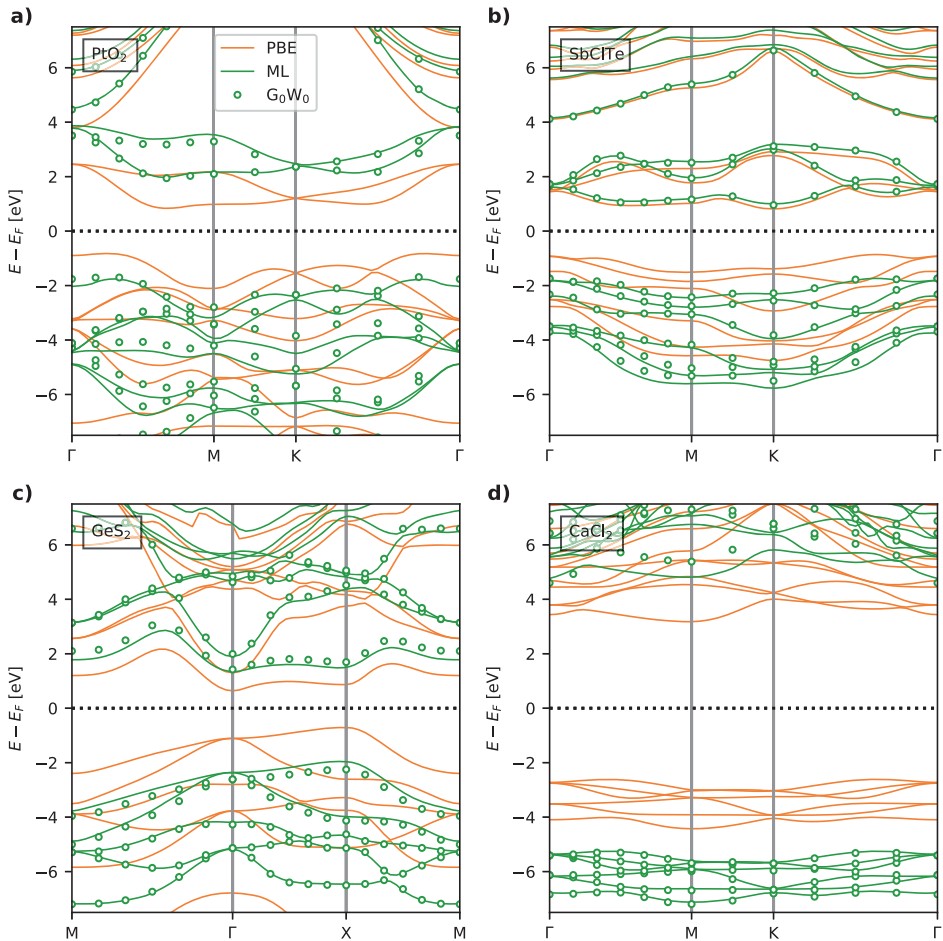

**Fig. 5 Machine-learned band structures.** Examples of band structures for four 2D materials from the test set. Both PBE and GW band structures are shown along with the ML predictions. The materials are selected to cover a wide range in the prediction accuracy of the test set. Band structures for $PtO_2$ (**a**), SbClTe (**b**), $GeS_2$ (**c**), and $CaCl_2$ (**d**).

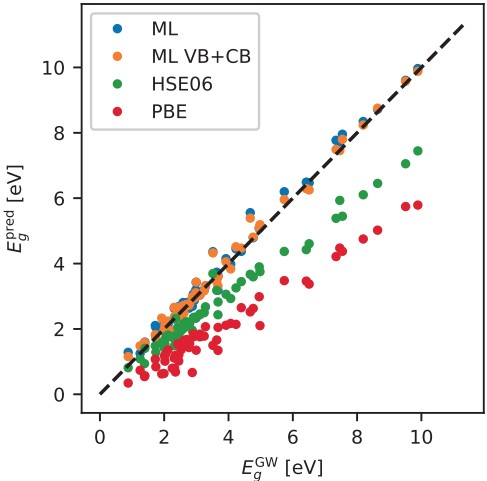

**Fig. 6 Comparison of bandgaps.** Parity plots for predicted bandgaps vs. GW bandgaps for PBE and HSE06 and two different ML models predicting GW corrections for either all bands (MAE = 0.18 eV) or only valence and conduction bands (MAE = 0.15 eV) which significantly outperform PBE and HSE06 with MAEs of 1.70 and 0.85 eV, respectively.

database of 270 $G_0W_0$ bandgaps, they obtained a root mean square error (RMSE) of 0.24 eV. Rajan et al. used a Gaussian process to predict $G_0W_0$ bandgaps of 2D MXene crystals with a fingerprint encoding atomic and structural properties of the MXenes[17]. Employing a training set of 76 $G_0W_0$ MXene bandgaps, they obtained an RMSE of 0.14 eV.

We stress that both the inorganic bulk semiconductors considered ref. [26] and, in particular, the MXene 2D crystals of ref. [17], represent more homogeneous sets of materials than the 2D crystals considered in the present work. Nevertheless, with an RMSE of 0.26 and 0.21 eV on the predicted $G_0W_0$ bandgap for the models trained on 8VB + 4CB and VB + CB, respectively, our general ML model with purely electronic fingerprints, is comparable in accuracy to the more system-specific ML models.

Additionally, by applying our ML model on ∼700 semiconductors from C2DB we have found the bandgap to change nature (direct/indirect) in 12% of the materials when comparing the PBE and ML bandgaps. For these materials, 72% shift from direct to indirect gaps.

**Effective masses.** Since the ML model can be used to calculate $G_0W_0$ energies at any $k$-point grid, it is possible to use the method to calculate effective masses. Effective masses at the valence and conduction band extrema can be calculated by fitting a second-

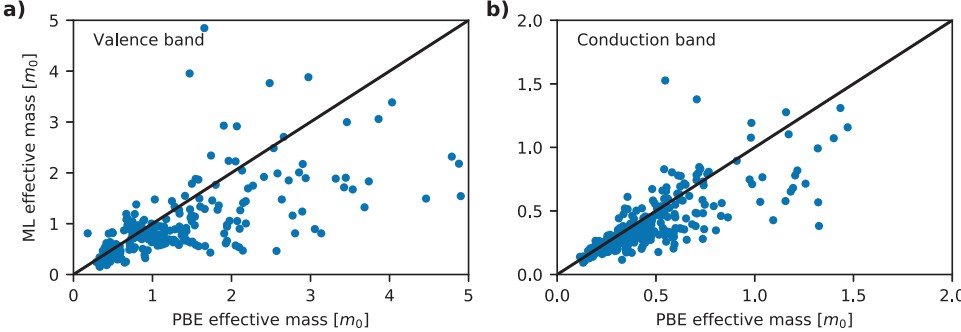

**Fig. 7 Effective masses.** Comparison of effective masses calculated using PBE and ML-$G_0W_0$ eigenvalues for valence and conduction band of ∼800 materials. **a** Shows effective masses for the valence bands and **b** shows for the conduction bands. There seems to be a (weak) systematic trend for the ML model to predict smaller effective masses than PBE for both valence and conduction bands.

order polynomial to the energies at a densely sampled $k$-point grid centered around the band extrema[20,21]. This method is generally challenging with $G_0W_0$ due to the high computational cost of calculating the energies at sufficiently dense $k$-point grids, but using the ML model it is possible to achieve accurate estimates of the $G_0W_0$ effective masses.

Figure 7 shows effective masses calculated using PBE and ML energies for ≈330 materials using a $k$-point density of 55/Å$^{-1}$ in a radius of 0.16 Å$^{-1}$.

The validity of the polynomial fit is evaluated using a mean absolute relative error (MARE) metric. The MARE is defined as the absolute difference between the parabolic fit and the actual ML-$G_0W_0$ band energies averaged over an energy range of 100 meV (from the band extremum) relative to the actual band energies averaged over the same energy range. The data shown in Fig. 7 includes only fits with MARE less than 10%.

Returning to Fig. 7 we note that the effective masses obtained with ML-$G_0W_0$ can deviate quite significantly from the PBE values. Specifically, the mean absolute deviation is 0.31$m_0$ and 0.19$m_0$ for valence and conduction bands, respectively, corresponding to relative deviations of 32 and 28%. We can also deduce that the ML-$G_0W_0$ method has a general tendency to yield smaller effective masses than PBE, although deviations from this trend occur relatively often.

## Discussion

**Feature importance**. Often the evaluation of a machine learning model stops after considering the overall performance in terms of an objective function like the MAE. However, important insight may be gained by analysing how the model responds to different features in the input data. This is particularly important when devising new types of fingerprints. To extract information about the role of the different features composing the fingerprint vectors used in the present work, a feature importance analysis is performed using a feature subset hold-out method. The features are grouped at two different levels: The first level has four groups, namely the RAD-PDOS components, the ENDOME components, the extra features covering the PBE gap, occupation number, distance to the Fermi level, and finally the in-plane and out-of-plane polarizabilities. The second level breaks the RAD-PDOS and ENDOME components further down into their individual $ll'$ angular momentum blocks and operator matrix elements, respectively. The analysis is carried out in two complementary ways where a group of features is either used exclusively or dropped from the full fingerprint when training the ML model.

Figure 8 shows the test set MAE on individual state energies for the various feature groups with the all-feature baseline indicated by the vertical black line. Focusing first on panel (a), the analysis shows that both the RAD-PDOS and ENDOME perform well by

themselves, though not as well as the full fingerprint. The extra features, in particular the polarisabilities, are unable to produce an accurate ML model. The poor performance of the polarisability-only feature is unsurprising as this feature is fully material-specific and not even able to distinguish between occupied and unoccupied states. Panel (b) shows the same analysis when the feature groups are broken further down. When used alone, the $pp$, $ss$, and $sp$ components of the RAD-PDOS perform best followed by the various operator matrix elements of the ENDOME. An interesting observation is that at this level of feature grouping, almost any group of features can be dropped without increasing the MAE, except for the in-plane polarisability, $\alpha_{xy}$, which results in a significant 27% increase of the MAE from 0.11 to 0.14 eV. This reveals a clear feature synergy since $\alpha_{xy}$ in itself does not have any predictive ability unless it is combined with other features (see below). In general, there seems to be some redundant information in the various fingerprint components since dropping any of the feature sets, at least at the second level of grouping, does not affect the test score by much. In some cases, the model might even gain performance when dropping some features (not visible on the scale of the plot). This suggests that a feature selection algorithm prior to the prediction algorithm might in general slightly improve the performance of the model. However, since gradient boosting algorithms like XGBoost already has some implicit feature selection in the training iterations, the improvement is not expected to be significant and is thus not considered here.

**SHAP analysis**. The role of the $\alpha_{xy}$ feature and its synergy with other features is further investigated using the general feature importance method SHAP, which is a game-theoretic approach to explain the output of any machine learning model[27]. SHAP builds an explanation model on top of an ML model which relates the output from the ML model to the importance of individual features for each predicted output. The SHAP values for a given feature can thus be interpreted as the direct effect of that feature on the model output, i.e. the difference between the model's prediction when used with and without that particular feature in the input. Figure 9a shows the SHAP values for $\alpha_{xy}$ as a function of $\alpha_{xy}$. Only states from the test set are shown in Fig. 9, and the color code in panel (a) reflects the occupancy of the state. The plot shows a surprisingly clear trend: The SHAP values for occupied states increase consistently and monotonously for increasing $\alpha_{xy}$ while the opposite trend is seen for the empty states. In the following, we present a physical explanation for this observation.

The $G_0W_0$ correction can be split into two terms with distinctly different physical origin: $\Delta E_{nk}^{QP} = (v_{nk}^x - v_{nk}^{xc}) + \Delta_{nk}^{scr}$. The first term (in parenthesis) represents the difference between the local xc-potential (in this case the PBE potential) and the nonlocal exact exchange potential while the last term accounts for the interaction of

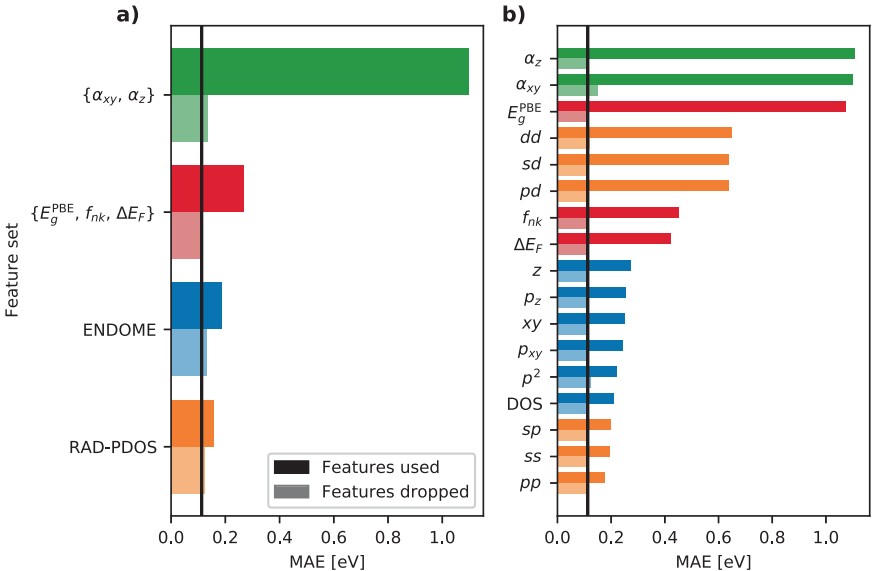

**Fig. 8 Feature analysis of ML model.** Solid bars refer to an ML model using only the specific features while the shaded bars are for an ML model without these features. **a** High-level feature group. **b** Low-level feature groups.

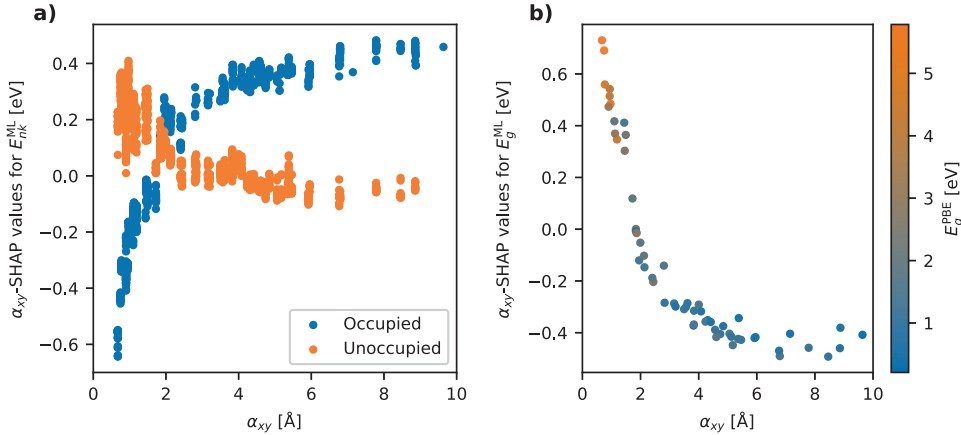

**Fig. 9 SHAP analysis. a** SHAP values for $\alpha_{xy}$ for the prediction of GW correction energies color-coded by occupancy. For materials with a low polarisability, the ML model predicts a more negative GW correction for the occupied states and a more positive correction for the unoccupied states. For materials with a high polarisability, the occupied states are predicted with a more positive correction when using the polarisability as a feature while the unoccupied states are only weakly affected. **b** SHAP values for $\alpha_{xy}$ for the prediction of bandgaps. This shows that the bandgap increases for materials with a low $\alpha_{xy}$ and decreases for high $\alpha_{xy}$ values.

the electron/hole with its own polarization cloud. The first term is typically negative for occupied states and positive for unoccupied states (Hartree–Fock typically opens the PBE gap), but its magnitude depends on the detailed shape of the wave functions of the system. In particular, this term can be quite different for different states of the same material. Moreover, one does not expect the size of this term to correlate with the material's static polarisability and thus it should not be captured by the $\alpha_{xy}$-SHAP values. The second term is always positive for occupied states (hole quasiparticles) and negative for unoccupied states (electron quasiparticles) because the Coulomb interaction of the bare particle with its oppositely charged polarization cloud will always stabilize the quasiparticle, thus shifting occupied states up and empty states down in energy[28–30]. Now, the shape and size of the polarization cloud does not depend on the detailed shape of the wave function but is largely governed by the (microscopic) polarisability of the material. Therefore, on purely physical grounds, the static macroscopic polarisability, $\alpha_{xy}$, is expected to provide a good descriptor for $\Delta_{nk}^{scr}$: A large value of $\alpha_{xy}$ signals high screening ability

of the material and therefore large QP polarization clouds, which in turn will yield a large $\Delta_{nk}^{scr}$ (with opposite signs for occupied/empty states). This is exactly what is seen in Fig. 9a. By subtracting the $\alpha_{xy}$-SHAP values for the states at the CBM and VBM, we obtain the $\alpha_{xy}$-SHAP values for the bandgap correction, see Fig. 9b. These show that the $\alpha_{xy}$ feature increases the bandgap in materials with low screening and decreases the bandgap in materials with high screening. Again, this is perfectly in line with the physical understanding of screening-induced renormalization of the bandgaps[28–30].

It can be noted that the $\alpha_{xy}$-SHAP values for the state energies and bandgaps are significantly larger than the change in the MAE upon including/dropping $\alpha_{xy}$ from the feature set, see Fig. 8b. For example, the $\alpha_{xy}$-SHAP values for the bandgap range from $-0.50$ to $0.70$ eV while the MAE decreases by $0.03$ eV when $\alpha_{xy}$ is included. This is due to the redundant information carried by the feature set. When the model is trained without $\alpha_{xy}$ as a feature, other features can, to a large extent, provide the same information. For example, the PBE bandgap alone correlates fairly well with $\alpha_{xy}$. To test this

hypothesis, we have carried out the same SHAP analysis for $E_g^{PBE}$ on a model trained with and without $\alpha_{xy}$ in the feature set. The analysis shows that when $\alpha_{xy}$ is used to train the model, the $E_g^{PBE}$-SHAP values are fairly low (below $\pm 0.1$ eV) and do not show any clear trends. In contrast, when $\alpha_{xy}$ is not included in the fingerprint, the $E_g^{PBE}$-SHAP values are very similar to the $\alpha_{xy}$-SHAP values shown in Fig. 9, although the values are slightly smaller and the trend less pronounced. This shows that in the absence of $\alpha_{xy}$ the model uses $E_g^{PBE}$ to encode similar information. However, the model also finds that $\alpha_{xy}$ provides a better description of $\Delta_{nk}^{scr}$ than does $E_g^{PBE}$, which is why the SHAP values of $E_g^{PBE}$ are dwarfed by those of $\alpha_{xy}$ when both features are available for learning.

**Summary**. In summary, we have introduced two different methods to generate fingerprints of individual electronic states based on information available from a standard DFT ground-state calculation (eigenvalues and wave functions). The fingerprints were used to train a decision-tree-based ML model to predict the $G_0W_0$ corrections to the PBE band structure of a 2D semiconductor. The model achieves an MAE of 0.14 eV for individual state energies, which is reduced to 0.11 eV when the static polarisability is included in the fingerprint. For the band-gap, the MAE is 0.15–0.23 eV depending on whether the model is trained on all bands or only the valence/conduction bands and whether or not the static polarisability is included in the fingerprint. This level of precision is highly encouraging considering that the noise on the employed $G_0W_0$ data for individual state energies could be on the order of 0.05 eV and that the accuracy of the $G_0W_0$ method itself, when evaluated against experimental bandgaps, is about 0.3 eV. Since the bottleneck of the computations is the self-consistent DFT calculation (in particular the structural relaxation if performed), the method enables GW-quality band structures at the cost of a DFT calculation. Although the current work has focused on states in periodic 2D crystals, the methods can be straightforwardly used to fingerprint states in 3D crystals as well as non-periodic structures like molecules or surfaces. While the fingerprint methods can be used for e.g., 3D crystals, the ML model trained on 2D materials will not be transferable since some of the fingerprint components are divided into in-plane and out-of-plane parts. To use the full method of fingerprints and ML model for 3D crystals would require an ML model trained on a database of GW calculations of such systems.

## Methods
This section describes the definition and generation of the Energy Decomposed Operator Matrix Elements (ENDOME) and Radially Decomposed Projected Density Of States (RAD-PDOS) fingerprints. In addition, the $G_0W_0$ band structure data set is presented along with a description of the employed machine learning model.

**Electronic state fingerprints**. The ENDOME fingerprint is based on operator matrix elements between electronic states (here assumed to be Bloch states of a periodic crystal)

$$A_{nk,n'k'} = |\langle nk|\hat{A}|n'k'\rangle|^2 \qquad (1)$$

where $\hat{A}$ is some operator. For a reference state $|nk\rangle$ with energy $\varepsilon_{nk}$, the ENDOME fingerprint is defined as

$$m_{nk}^A(E) = \sum_{n'k'} A_{nk,n'k'} G\big(E - (\varepsilon_{nk} - \varepsilon_{n'k'}); \delta_E\big) \exp(-\alpha_E E) \, \text{sign}(E_F - \varepsilon_{n'k'}), \qquad (2)$$

where $G(x; \delta)$ is a Gaussian of width $\delta$ centered at $x = 0$. This function encodes the matrix element between the reference state and all other states at an energy distance of $E$ from the reference state. In principle, any operator can be used to create fingerprints, but in this study, we include the position operators ($x, y, z$), the momentum operators ($\nabla_x, \nabla_y, \nabla_z$), and the Laplace operator ($\nabla^2$). These operators are all diagonal in the $k$ index. In addition, we include the all-one matrix, $A_{nk,n'k'} = 1$, which essentially yields the density of states (DOS) translated to the energy of the reference state, $\varepsilon_{nk}$.

In practice, the function $m_{nk}^A(E)$ is represented on a uniformly spaced energy grid with 50 energy points from $-10$ to 10 eV around the reference state. Since we consider 2D materials, the in-plane ($x$ and $y$) components of both the position and momentum operators are collected into a single fingerprint vector (i.e., $m_{nk}^{xy} = m_{nk}^x + m_{nk}^y$ and similarly for the momentum operator) while the out-of-plane $z$ component is treated separately. For a given reference state, the ENDOME fingerprint thus consists of six 50-dimensional vectors resulting in a total of 300 features.

The RAD-PDOS encodes the electronic structure in terms of the density of states projected onto atomic orbitals. Specifically, a correlation function in energy and radial distance is defined as

$$\rho_{nk}^{\nu\nu'}(E, R) = \frac{1}{N_e} \sum_{n'k'aa'} \rho_{nk}^{a\nu} \rho_{n'k'}^{a'\nu'} G\big(R - |R_a - R_{a'}|; \delta_R\big) \exp(-\alpha_R R) G\big(E - (\varepsilon_{nk} - \varepsilon_{n'k'}); \delta_E\big)$$
$$\times \exp(-\alpha_E E) \, \text{sign}(E_F - \varepsilon_{n'k'}) \qquad (3)$$

where $N_e$ is the number of electrons in the system, $a$ and $a'$ denote atoms in the primitive unit cell and the entire crystal, respectively, and $\nu$ and $\nu'$ denote atomic orbitals. The atomic projections are given by

$$\rho_{nk}^{a\nu} = |\langle \psi_{nk}|a\nu\rangle|^2 \qquad (4)$$

The functions $\rho_{nk}^{\nu\nu'}(E, R)$ are represented on a uniform $(E, R)$-grid of size $25 \times 20$ spanning the intervals from $-10$ to 10 eV (centered around the reference energy $\varepsilon_{nk}$) and 0 to 5 Å, respectively. For the Gaussian smearing functions we use $\delta_E = 0.3$ eV and $\delta_R = 0.25$ Å, respectively. For a given state, the RAD-PDOS fingerprint consists of six 2D grids of 500 points each resulting in a total of 3000 features.

Figure 2 shows examples of ENDOME and RAD-PDOS fingerprints for three different states at the $K$-point of $MoS_2$. Note that some of the RAD-PDOS fingerprints are qualitatively similar (e.g., sp and pp) but the scales differ by about an order of magnitude. This is due to the fact that the density of states projected onto $s$ and $p$ orbitals have a similar dependence on energy.

**The $G_0W_0$ data set**. The data set comprises quasiparticle (QP) energies from 286 $G_0W_0$ band structures of non-magnetic 2D semiconductors covering 14 different crystal structures and 52 chemical elements. The QP energies have been obtained from plane-wave-based one-shot $G_0W_0$@PBE calculations with full frequency integration and were produced as a part of the Computational 2D Materials Database (C2DB)[20,21]. The data set has been described and analysed in detail in ref. [25].

The QP energies of the data set have been calculated under the standard assumption that the $G_0W_0$ self-energy can be treated within first-order perturbation theory and linearized around the non-interacting reference energy, $\omega = \varepsilon_{nk}$, leading to the expression

$$E_{nk}^{QP} \approx \epsilon_{nk} + Z\text{Re}\big[\langle \psi_{nk}|\Sigma(\epsilon_{nk})|\psi_{nk}\rangle\big] \qquad (5)$$

where

$$Z = \left(1 - \frac{\partial \Sigma}{\partial \omega}\bigg|_{\omega = \epsilon_{nk}}\right)^{-1} \qquad (6)$$

is the QP weight and $\psi_{nk}$ is the PBE wave function with eigenvalues $\epsilon_{nk}$. In practice, the $G_0W_0$ correction to the PBE energies, $\Delta E_{nk}^{QP} = E_{nk}^{QP} - \epsilon_{nk}$, were used as targets for the machine learning model.

To ensure the highest data quality, the original data set was filtered such that only states with QP weights between 0.7 and 1.0 were kept. As shown in ref. [25] the MAE on the QP correction of such states due to the linearization of the QP equation is 0.04 eV.

**Machine learning model**. The choice of learning algorithm for a machine-learned model depends on different considerations such as the amount of training data available and the nature of the learning objective (regression/classification, discrete/continuous). The fingerprints presented here are not designed for a specific learning algorithm and can thus be used to train a wide range of algorithms. For this specific purpose of predicting $G_0W_0$ QP energies, several types of algorithms including tree-based ensemble methods, neural networks, and Gaussian process regression have been considered and tested. The machine learning model is built using a gradient boosting method from the XGBoost distribution based on decision trees in an ensemble[24]. The choice of XGBoost as a learning algorithm is based on its generality and good performance across multiple machine learning applications, the possibility to extract knowledge from single features, and the ability of training on large amounts of data. For this specific purpose, a neural network and a gaussian process regression method have also been tested resulting in similar prediction accuracy.

A train and test set is created using a random 80/20% split on the material level which results in a train set of 228 materials (37851 QP energies) and a test set of 58 materials (8766 QP energies). Hyperparameters of the learning algorithm (max depth = 5, learning rate = 0.15, and number of estimators = 60) are tuned using a grid search method with fivefold cross-validation of the 80% train set. The performance of the machine learning is based on the mean absolute error (MAE) of the 20% test set.

Since the test set size is only 58 materials, the test MAE might exhibit some test set dependence. To evaluate this effect, the entire process of splitting the data in 80/20% train/test set, training the model using fivefold cross-validation on the train set, and evaluating the MAE of the test set, has been repeated 100 times using different seeds for the random split. The distribution of the 100 test MAEs have a mean of 0.13 eV and a standard deviation of 0.02 eV. We note that the specific test set used for Table 1 yields an MAE within one standard deviation from the mean.

Since the XGBoost model is based on decision trees some small discontinuities in-band energies might be introduced by the model. When calculating effective masses using a harmonic fit on a much smaller energy scale than the full band structures it was necessary to use a neural network (feed-forward network with three hidden layers with 200 neurons and tanh activation functions) to ensure a more continuous output. This NN yielded a test MAE of 0.13 eV compared to the 0.11 eV of the XGBoost model.

## Data availability
The structures of the materials used in this study have been deposited in C2DB[22] (https://doi.org/10.11583/DTU.14616660.v1). The data set generated for this study is available at https://gitlab.com/knosgaard/electronic-structure-fingerprints.

## Code availability
The Python code used to compute the fingerprints can be found here https://gitlab.com/knosgaard/electronic-structure-fingerprints.

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

## Acknowledgements
The Center for Nanostructured Graphene (CNG) is sponsored by The Danish National Research Foundation (project DNRF103). We acknowledge funding from the European Research Council (ERC) under the European Union's Horizon 2020 research and innovation program Grant No. 773122 (LIMA) and Grant agreement No. 951786 (NOMAD CoE). K.S.T. is a Villum Investigator supported by VILLUM FONDEN (grant no. 37789).

## Author contributions
N.R.K. and K.S.T. developed the initial concept. N.R.K. developed the Python code for computing the fingerprints and training the machine learning models. K.S.T. supervised the work and helped in the interpretation of the results. All authors modified and discussed the paper together.

## Competing interests
The authors declare no competing interests.

## Additional information

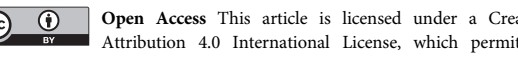

