## [Peer Review File · Nature Communications]

Representing individual electronic states for machine learning
GW band structures of 2D materialsREVIEWER COMMENTS

Reviewer #1 (Remarks to the Author):

This article reports on the development of a novel machine-learning (ML) framework to predict electronic band structures at the GOWO level. The authors propose a fingerprint (a.k.a. descriptor) based on a "energy-decomposed operator matrix elements" and a "radially-decomposed projected density of states", in addition to other quantities such as the electronic polarisability. The (ML) algorithm learns, on a remarkably small training set, how to map a DFT band structure into a GOWO band structure with relatively good accuracy. The framework is inspected with clustering techniques and feature importance analysis, providing insights on the model success.

The manuscript is well written, easy to follow and with clear supporting figures. I just note here a couple of mistakes: 1) at the first page there is a missing reference "[?]" right after the word "GOWO@LDA" and 2) in section 4.3 there is a forgotten comment "XXX: should we mention what they are? ..."

The results are very noteworthy, in particular this novel approach allows to compute the full band structure with a ML algorithm, at variance with earlier approaches who focused on the band gap only. The methodology is sound and the methods clearly outlined, I appreciate in particular the effort on understanding the behaviour of the descriptor and of the algorithm through importance analysis.

I think this work will have an important impact on the field, in particular I find two very relevant aspects. On the one hand, this work will equip high-throughput studies with much more accurate band structures at a reasonable computational cost. Database of GW calculations for training will become more populated over time, potentially increasing the accuracy of this method to the level of a direct GW calculation. On the other hand this work has shed some light on what are the important ingredients to craft an effective electronic structure descriptor, where I refer in particular to the discussion of Fig. 7.

I recommend publication of this manuscript without any substantial revision.

Reviewer #2 (Remarks to the Author):

The authors created a new method to predict a material's k-resolved band structure with GW level accuracy. They train their new model using a reasonably large dataset of 2D band structures and supplement their descriptor set with a few additional PBE-level obtained input parameters. They show their approach can reliably estimate the band structure while also providing a detailed analysis using tSNE and SHAP feature analysis. Overall, this work is a nice demonstration of using machine learning to reduce computationally expensive calculations to more accessible levels. There is also value in obtaining k-resolved information, as the authors note in their introduction.

However, I am not sure this paper would be of interest to the general readership of Nat. Comm. with the current content. I think the manuscript would be a better fit for a specialized journal like npj Computational Materials. I believe a few things the authors could do to tailor the paper to a more general journal.

1. The model was trained on 2D materials, and at the end of the paper (page 5), the authors note, "the methods can be straightforwardly used to fingerprint states in 3D crystals as well as non-periodic structures like molecules or surfaces." It would be nice if they provided a demonstration of this claim? Proving this goal would significantly increase the broad interest in their technique. In fact, the choice of the hold-out band structure is also a bit odd: are PtO₂, SbClTe, GeS₂, and CaCl₂ meaningful 2D materials? Or are these bulk structures? This point is not very clear.

2. The authors could also provide a better comparison of their model accuracy relative to the literature. For example, they highlight that their model MAE is 0.18 eV at the end of the results section. How does this compare to other ML models?

3. The authors should create a few different random states for their hold-out test set. This is a relatively small composition space (only 286 compounds), and the 58 compounds in the test set can massively influence the model's accuracy. I would recommend the authors analyze a few test sets and report the average MAE.

4. At the top of page 5, the authors mention that "...feature selection might improve the model's performance further." This is also valuable to reduce the risk of overfitting. Is there a reason why feature selection is not performed?

5. Why is α_{xy} so critically vital to the model but α_z not nearly as important?

6. XGBoost is powerful, but a short discussion of why they chose XGB over gradient boosting or random forest, for example, could be useful.

7. How did they settle on the density of the grid for the k-resolved training data? Were other meshes considered?

8. The author's reason for wanting band structure was to provide information on the type of transition (direct/indirect) or electron/hole mass. They could compare these properties obtained from their model as well as PBE and GW methods. That could provide further support for their model.

Finally, a minor note - the authors should check section 4.3; they left a note to themselves in the submitted text that should not be there...

Reply to reviewer #2

We thank the reviewer for taking the time to read our manuscript and for the valuable suggestions for improvements. Except for one point (the first one below), we have followed all your suggestions, and we hope that you can now recommend our paper for publication.

Reviewer comment:

The model was trained on 2D materials, and at the end of the paper (page 5), the authors note, "the methods can be straightforwardly used to fingerprint states in 3D crystals as well as non-periodic structures like molecules or surfaces." It would be nice if they provided a demonstration of this claim? Proving this goal would significantly increase the broad interest in their technique. In fact, the choice of the hold-out band structure is also a bit odd: are PtO₂, SbClTe, GeS₂, and CaCl₂ meaningful 2D materials? Or are these bulk structures? This point is not very clear.

Reply:

The model was exclusively trained on and applied to 2D materials, that is atomically thin 2D structures surrounded by vacuum. The materials PtO₂, SbClTe, GeS₂, and CaCl₂ used as examples are all 2D materials. No bulk materials were considered in the current work. The machine learning model trained on 2D materials cannot be directly used for bulk 3D structures because some components of the fingerprints are designed for 2D materials (e.g. we average over the in-plane directions x and y while treating the out-of-plane z -direction separately). However, it would be very straightforward to construct similar fingerprints suitable for 0D, 1D or 3D structures. The reason why this has not been done in the current work is that there are no available GW band structure databases for such materials. Having said that we agree that this is a very relevant point and we are currently working to create GW databases that could serve as training sets for further machine learning studies. This is, however, a very significant endeavour that will take at least one year and is outside the scope of the current work. This is now made more clear in the paper, see highlighted red text on page 6.

Reviewer comment:

The authors could also provide a better comparison of their model accuracy relative to the literature. For example, they highlight that their model MAE is 0.18 eV at the end of the results section. How does this compare to other ML models?

Reply:

We have included a new paragraph at the end of the results section (just before the new paragraph on effective masses) where we make a more direct comparison of our model's performance against two previously published methods where nonlinear regression was used to predict G₀W₀ band gaps of bulk semiconductors and 2D MXene crystals, respectively. This comparison shows that our ML model is at least on par with the previous models while at the same time being more general. We do, however, stress that the main novelty of our method is that it allows for prediction of state-specific properties such as individual band energies and not just material specific properties such as the band gap.

Reviewer comment:

The authors should create a few different random states for their hold-out test set. This is a relatively small composition space (only 286 compounds), and the 58 compounds in the test set can massively influence the model's accuracy. I would recommend the authors analyze a few test sets and report the average MAE.

Reply:

We have repeated the process of splitting the data in 80/20 % train/test, training the model using 5-fold cross-validation and computing the test MAE a total of 100 times to estimate this effect. The analysis shows that our reported test MAE is within one standard deviation of the test set MAEs and therefore we assume the selected test set to be fair and representative. To make this clear we have added a new paragraph explaining the multiple test set analysis in section 4.3 (“Machine learning model”).

Reviewer comment:

At the top of page 5, the authors mention that "...feature selection might improve the model's performance further." This is also valuable to reduce the risk of overfitting. Is there a reason why feature selection is not performed?

Reply:

A gradient boosting algorithm like XGBoost has some implicit feature selection in the training iterations, and therefore the improvement by using a separate feature selection algorithm is likely to be very small and therefore it is not performed here. But for a general machine learning model there might be something to gain by performing feature selection prior to training the actual prediction model, and thus it cannot be ruled out that feature selection might have an effect. We now mention this explicitly at the end of the second paragraph of the “Discussion” section.

Reviewer comment:

Why is α_{xy} so critically vital to the model but α_z not nearly as important?

Reply:

The reason is simply that the in-plane (x and y) components of the polarizability are significantly larger than the out-of-plane (z) component for any 2D material. In fact, the former are often 1-2 orders of magnitude larger than the latter. The physical reason is that the electrons are confined in the out-of-plane direction and therefore not able to provide much screening in response to a field polarized in that direction. This means that the strength of the screened Coulomb interaction, which is a vital ingredient of the GW self-energy, is mainly determined by α_{xy} . Note, however, that this will not be true for a bulk material where all three components of α are expected to be equally important.

Reviewer comment:

XGBoost is powerful, but a short discussion of why they chose XGB over gradient boosting or random forest, for example, could be useful.

Reply:

XGBoost is very closely related to other gradient boosting algorithms or ensemble methods like random forest. XGBoost is chosen because of its general performance across multiple use cases, but we have also tested a gaussian process regression (GPR) method which showed similar scores. XGBoost has the advantage of being able to handle large amounts of data compared to GPR.

Reviewer comment:

How did they settle on the density of the grid for the k-resolved training data? Were other meshes considered?

Reply:

The density of the k-grid for the training data was chosen as a balance between accuracy and computational cost. A detailed description of the GW calculations including convergence tests for the k-point grids can be found in 2D Materials 5, 042002 (2018).

Reviewer comment:

The author's reason for wanting band structure was to provide information on the type of transition (direct/indirect) or electron/hole mass. They could compare these properties obtained from their model as well as PBE and GW methods. That could provide further support for their model.

Reply:

We have calculated effective masses using the ML method for 800 materials. This shows a tendency of the ML model to give smaller effective masses compared to PBE. Additionally, we have calculated band structures of 700 non-magnetic semi-conductors from C2DB. These band structures will be published on the C2DB web database. From these we find that applying the ML model results in a shift in band gap characteristics (direct/indirect) in 12 % of the materials, with 72 % of the cases being shifts from direct to indirect band gaps.

These new results required a quite substantial amount of extra work and is the main reason for the long delay in our resubmission. The new results are discussed in the new subsections “Band gaps” and “Effective masses” at the end of the Results section.

REVIEWERS' COMMENTS

Reviewer #2 (Remarks to the Author):

The authors have addressed each of the concerns adequately. Their paper is now scientifically sound and provides some reasonably noteworthy results. The changes have also provided a better comparison with the current literature.

I would have preferred if the authors also modified the title of their article to reflect that this is related to 2D materials. But that is a minor point. I would recommend this paper for publication.

Reply to reviewer #2

We thank the reviewer for taking the time to read our manuscript and for the valuable suggestions for improvements. We have followed the reviewers suggestions.

Reviewer comment:

The authors have addressed each of the concerns adequately. Their paper is now scientifically sound and provides some reasonably noteworthy results. The changes have also provided a better comparison with the current literature.

I would have preferred if the authors also modified the title of their article to reflect that this is related to 2D materials. But that is a minor point. I would recommend this paper for publication.

Reply:

We have updated the title to “Representing individual electronic states for machine learning GW band structures of 2D materials”.